# Intend to Move: A Multimodal Dataset for Intention-Aware Human Motion Understanding

**Ryo Umagami**
The University of Tokyo
umagami@mi.t.u-tokyo.ac.jp

**Liu Yue**
The University of Tokyo
liuyue@mi.t.u-tokyo.ac.jp

**Xuangeng Chu**
The University of Tokyo
xuangeng.chu@mi.t.u-tokyo.ac.jp

**Ryuto Fukushima**
The University of Tokyo
fukushima@mi.t.u-tokyo.ac.jp

**Tetsuya Narita**
The University of Tokyo
narita@mi.t.u-tokyo.ac.jp

**Yusuke Mukuta**
The University of Tokyo
RIKEN AIP
mukuta@mi.t.u-tokyo.ac.jp

**Tomoyuki Takahata**
Tokyo Denki University
t-tkht@mail.dendai.ac.jp

**Jianfei Yang**
Nanyang Technological University
jianfei.yang@ntu.edu.sg

**Tatsuya Harada**
The University of Tokyo
RIKEN AIP
harada@mi.t.u-tokyo.ac.jp

## Abstract

Human motion is inherently intentional, yet most motion modeling paradigms focus on low-level kinematics, overlooking the semantic and causal factors that drive behavior. Existing datasets further limit progress: they capture short, decontextualized actions in static scenes, providing little grounding for embodied reasoning. To address these limitations, we introduce *Intend to Move (I2M)*, a large-scale, multimodal dataset for intention-grounded motion modeling. I2M contains 10.1 hours of two-person 3D motion sequences recorded in dynamic realistic home environments, accompanied by multi-view RGB-D video, 3D scene geometry, and language annotations of each participant's evolving intentions. Benchmark experiments reveal a fundamental gap in current motion models: they fail to translate high-level goals into physically and socially coherent motion. I2M thus serves not only as a dataset but as a benchmark for embodied intelligence, enabling research on models that can reason about, predict, and act upon the "why" behind human motion. Data and code are available at https://ummaaa.github.io/intend-to-move.

## 1 Introduction

To endow embodied AI with the ability to understand, predict, and seamlessly interact with humans, we must first teach it the fundamental grammar of human behavior: intention. Humans do not move

39th Conference on Neural Information Processing Systems (NeurIPS 2025) Track on Datasets and Benchmarks.

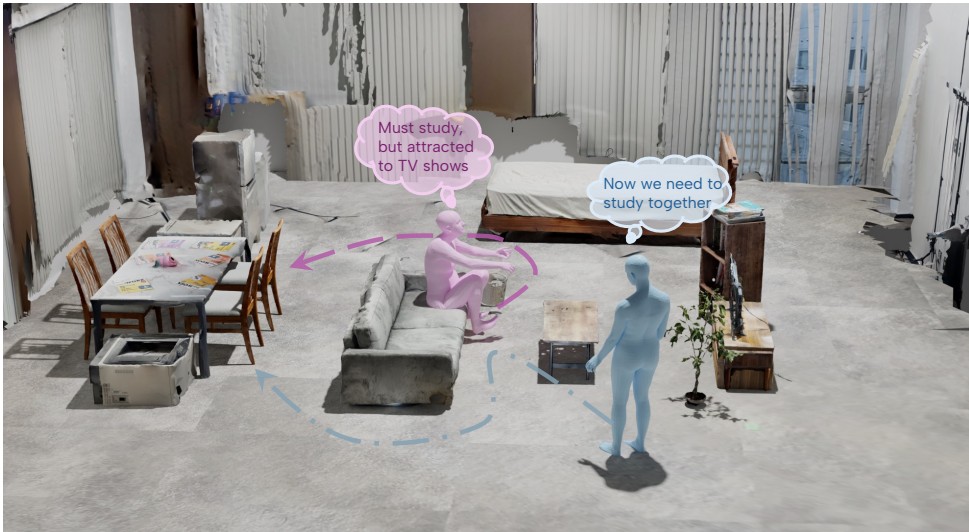

Figure 1: **Dataset overview.** Our dataset, I2M, captures complex, intention-driven human motion in realistic home environments, featuring rich interactions both between humans and with their surrounding scene.

in a vacuum; our actions are purposeful, driven by a complex interplay of goals, environmental context, and social dynamics. As Figure 1 illustrates, even a simple domestic scene is a stage for a complex web of intersecting intentions. Anticipating human actions in everyday scenarios represents a critical frontier for embodied AI applications, from assistive robots that collaborate with humans in daily tasks to intelligent environments that adapt to human needs.

However, existing human motion datasets, while foundational, have largely focused on the kinematics of what a person does, not why. Seminal datasets such as Human3.6M [Ionescu et al., 2014] and CMUMocap capture isolated, short-term actions in sterile lab environments. While invaluable for kinematic modeling, they lack the semantic richness and contextual complexity necessary for embodied agents to reason about human goals and adapt their behaviors accordingly. Consequently, while current models are capable of predicting trajectories, they lack an understanding of the intentions driving those movements. This gap hinders the development of truly interactive, intention-aware embodied systems, including robots and virtual agents that act in accordance with human goals.

To bridge this gap, we introduce **I**ntend **to M**ove (I2M), a large-scale, multimodal dataset designed to catalyze research in intention-aware embodied motion modeling. I2M provides 10.1 hours of long-duration, two-person motion sequences captured in realistic, dynamic home environments. Each sequence is enriched with synchronized multi-view RGB-D video, detailed 3D scene geometry, and, most critically, fine-grained, timestamped natural language annotations of each individual's evolving intentions.

Our dataset is built on three core principles that advance embodied motion understanding beyond prior work:

- **Intention-Grounded Motion:** All captured motions are explicitly motivated by a hierarchy of goals, from long-term activities like "cleaning the house" to short-term needs like "getting a drink," enabling models to learn the causal link between intention and action.
- **Rich Human-Scene and Human-Human Interaction:** Every scenario involves two individuals interacting with each other and a cluttered, object-rich environment—crucial for modeling the social and environmental grounding that embodied AI systems must master.
- **Dynamic and Realistic Scenes:** Unlike static lab settings, our environments evolve over time, with objects moving and environmental states changing, challenging models to reason about adaptive, context-aware behavior in real-world settings.

To benchmark the challenges posed by I2M, we conduct extensive evaluations using a diffusion-based generative model. Our experiments reveal a fundamental limitation: while current models excel when

Table 1: **Summary of existing human motion datasets and ours.** "Modality" denotes the types of data provided in the dataset. "Annotation" denotes the semantic annotation (e.g., "object" for object category and pose). "MultiHuman" indicates whether multiple people appear in one clip. "Natural Clothing" indicates if subjects wear everyday clothes instead of mocap suits in RGB data. "Capture Method" indicates the method used to obtain the human motion.

| Dataset Name | Modality | Annotation | Multi Human | Natural Clothing | Capture Method | Hour | Frame | Seq |
|---|---|---|---|---|---|---|---|---|
| KIT WBHM [Mandery et al., 2015] | RGB | motion text | | | optical | 7.68 | | 3.7k |
| PiGraphs [Savva et al., 2016] | 3D scan | interaction graph | | | kinect | 2 | | 63 |
| PROX [Hassan et al., 2019] | RGB, 3D scan | | | ✓ | kinect | | 100k | 60 |
| i3DB [Monszpart et al., 2019] | RGB | object | | | manual | 2.4k | | 8 |
| GRAB [Taheri et al., 2020] | 3D scan | contact | | | optical | | 1.3m | 1.3k |
| MoGAZE [Kratzer et al., 2020] | 3D synthe | gaze, contact, object | | | optical | 3 | | |
| GTA-IM [Cao et al., 2020] | RGB-D synthe | | | | synthe | | 1m | 120 |
| HPS [Guzov et al., 2021] | RGB, 3D scan | | ✓ | (✓) | imu | | 300k | |
| SAMP [Hassan et al., 2021] | 3D synthe | | | | optical | 1.6 | 185k | |
| MultiDex [Li et al., 2022] | 3D synthe | contact | | | synthe | | 436k | 436k |
| COUCH [Zhang et al., 2022b] | RGB-D | contact | | | imu,kinect | 3 | | > 500 |
| HUMANISE [Wang et al., 2022] | 3D scan | motion text | | | synthe | | 1.2m | 19.6k |
| GIMO [Zheng et al., 2022] | 3D scan | gaze | | (✓) | imu | | 130k | 217 |
| RICH [Huang et al., 2022] | multi-view RGB, 3D scan | contact | | ✓ | pseudo-labeling | | 540k | 134 |
| BEHAVE [Bhatnagar et al., 2022] | multi-view RGB-D | contact, object | | ✓ | pseudo-labeling | | 15.2k | 321 |
| CIRCLE [Araújo et al., 2023] | 3D synthe | | | | optical | 10 | 4.3m | > 7k |
| Human-M3 [Fan et al., 2023] | multi-view RGB, 3D scan | | ✓ | ✓ | pseudo-labeling | | 12.2k | |
| JRDB-Pose [Vendrow et al., 2023] | panorama RGB | | ✓ | ✓ | manual | 1.1 | 57.7k | 54 |
| CHAIRS [Jiang et al., 2023] | multi-view RGB-D, 3D scan | object | | (✓) | optical, imu | 17.3 | | |
| HiK [Tanke et al., 2023] | multi-view RGB | object, motion category | ✓ | ✓ | pseudo + manual | 7.3 | | |
| TRUMANS [Jiang et al., 2024] | synthesized multi-view RGB-D | contact, motion text | | | optical | 15 | 1.6m | |
| InterCap [Huang et al., 2024] | multi-view RGB-D | | | ✓ | pseudo-labeling | | 67k | 223 |
| HOI-M$^3$ [Zhang et al., 2024] | multi-view RGB, 3D scan | object | ✓ | (✓) | imu | 20 | 180m | > 199 |
| Nymeria [Ma et al., 2024] | RGB+Grayscale, point cloud | motion text, gaze, traj | ✓ | | imu | 300 | 260m | 1200 |
| **I2M (Ours)** | multi-view RGB-D, 3D scan | intention text | ✓ | ✓ | optical | 10.1 | 8.73m | 215 |

conditioned on low-level cues like joint trajectories, they struggle to translate high-level semantic intentions into physically and socially plausible motion. This finding reveals a fundamental bottleneck in the development of embodied AI—the grounding of abstract goals into embodied physical behavior. By offering a rich, intention-centric dataset, I2M provides a foundation for advancing research toward embodied AI systems that can not only predict human motion, but also understand, anticipate, and act upon the underlying intent.

# 2 Related Work

Large-scale motion datasets are foundational for developing robust algorithms for human motion prediction, generation, and recognition. Pioneering works like AMASS [Mahmood et al., 2019] unified existing motion capture (mocap) data to power learning-based approaches. Subsequent efforts have expanded dataset scale and diversity by incorporating rich scene and object contexts [Jiang et al., 2024, Kim et al., 2024, Pan et al., 2023, Ma et al., 2024]. The quality of these datasets can be assessed across several dimensions, including their semantic richness, motion accuracy, and visual realism. Existing datasets have often excelled in one area at the expense of others. In this section, we review prior work through the lens of these dimensions and position our dataset, I2M, as a novel contribution that makes significant advances on all fronts.

## 2.1 Semantic Annotation: From Action Description to Intention

Understanding the meaning behind human movement requires rich semantic annotations. As summarized in Table 1, the nature of these annotations has evolved significantly over time. Early datasets provided sparse labels like motion categories [Ionescu et al., 2013, Guo et al., 2020, Ghorbani et al., 2021, Harvey et al., 2020, Shahroudy et al., 2016, Cai et al., 2022] or semantic attributes [Punnakkal et al., 2021]. While useful for classification, these labels lack the descriptive power to capture the nuances of complex actions. To address this, subsequent work introduced free-form text annotations that describe the physical execution of the motion itself. The KIT dataset [Plappert et al., 2016] was among the first to use full sentences, and more recent efforts like HumanML3D [Guo et al., 2022] and Motion-X [Lin et al., 2023] have provided large-scale, fine-grained textual descriptions. However,

these annotations do not explicitly capture the underlying goal or motivation driving the action. This is a critical limitation, as human behavior is not merely a sequence of movements but a series of actions performed to achieve a higher-level objective.

Our work makes a key contribution by annotating motions with intentions. Instead of describing the motion (e.g., "The person walks to the fridge, takes out a drink, and drinks it. "), we annotate the motivation (e.g., "The person is thirsty."). This shifts the focus from motion representation to goal-oriented reasoning. GIMO [Zheng et al., 2022] shares a similar motivation, leveraging gaze as an implicit proxy for intent. In contrast, our dataset provides explicit, text-based intentions, offering more direct and specific supervision. This enables a new class of problems where models must infer a sequence of plausible actions to satisfy a stated intention, rather than simply translating a motion description into kinematics.

## 2.2 The Challenge of Uniting Motion Accuracy and Realism

The fidelity of a motion dataset is determined by its capture methodology, which has historically presented a challenge in simultaneously achieving high motion accuracy and visual realism. Various methods have been developed, each with distinct advantages and disadvantages.

Optical motion capture systems are the gold standard for accuracy, providing high-fidelity kinematic data [Taheri et al., 2020, Mandery et al., 2015, Hassan et al., 2021, Araújo et al., 2023, Jiang et al., 2023, 2024, Kratzer et al., 2020]. However, most existing datasets captured with this method required subjects to wear specialized mocap suits, creating a significant visual domain gap from natural attire.

Other capture methodologies allow for the use of everyday clothes, but these methods present their own set of compromises. Vision-based systems—such as monocular [Lin et al., 2023, Cai et al., 2023, Riza Alp Gueler, 2018, Pavlakos et al., 2019, Kanazawa et al., 2019, Shimada et al., 2020, Rong et al., 2021, Goel et al., 2023, Ye et al., 2023, Luvizon et al., 2023], multi-view [Joo et al., 2015, 2017, Zhang et al., 2022a, Khirodkar et al., 2023, Grauman et al., 2024], and RGB-D tracking [Savva et al., 2016, Hassan et al., 2019, Fan et al., 2023, Huang et al., 2022, Bhatnagar et al., 2022, Huang et al., 2024, Tanke et al., 2023]—are accessible, but they tend to be less accurate and are susceptible to occlusions. Full-body inertial measurement units (IMUs) [Trumble et al., 2017, von Marcard et al., 2018, Kaufmann et al., 2021, Guzov et al., 2021, Zhang et al., 2022b, Zheng et al., 2022, Zhang et al., 2024, Jiang et al., 2023, Kaufmann et al., 2023, Lee and Joo, 2024, Yang et al., 2024, Kim et al., 2024, Cong et al., 2024, Ma et al., 2024] are another popular choice, offering robustness to occlusions while better preserving natural attire compared to mocap suits, but the motion data is typically less precise than optical systems due to issues like drift. Finally, while simulation offers scalability [Cai et al., 2021, Akada et al., 2022, Black et al., 2023, Jiang et al., 2024, Araújo et al., 2023, Li et al., 2024, Cao et al., 2020, Wang et al., 2022], it struggles to replicate the diversity and subtlety of real-world behavior.

This review highlights that existing datasets have generally necessitated a compromise between motion accuracy and visual realism. Our work, I2M, is among the first datasets to jointly provide both high-fidelity motion capture and visually realistic, natural-clothing RGB-D observations. As shown in Table 1, we provide a large-scale dataset that features both the high accuracy of an optical capture system and the visual realism of subjects in natural clothing. We achieve this by affixing small, visually unobtrusive markers to everyday garments (see Section A for details). This approach allows us to capture highly accurate motion and visually authentic RGB-D data simultaneously, providing a unique and valuable resource for the community.

In summary, I2M addresses both the semantic limitation of existing datasets, which overlook human intentions, and the methodological trade-off between motion accuracy and realism, offering a comprehensive foundation for intention-aware motion modeling.

## 3 I2M Dataset

### 3.1 Dataset Overview

The I2M dataset contains 10.1 hours of 3D human motion data across 215 temporally continuous sequences. The data was collected from 16 participants (9 males, 7 females), recruited to ensure a balanced distribution across age (20-50) and a variety of body shapes. The dataset is multimodal,

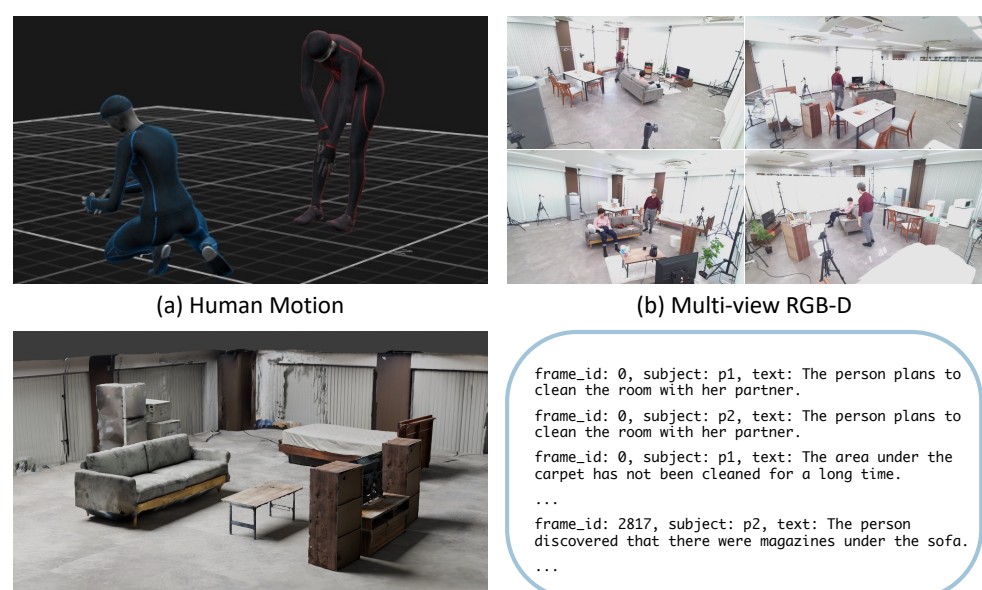

Figure 2: **Dataset modalities.** (a) Human motion sequence, (b) Multi-view RGB-D video, (c) 3D scene point cloud and mesh, and (d) Textual intention annotations.

Table 2: **Examples of the four categories of human intentions in our dataset.**

| Category | Intentions |
|---|---|
| Long-term Activities | leisure, studying, online meetings, entertaining friends, cleaning, exercising |
| Short-term Needs | drinking, eating, resting, going to the bathroom, turning on the TV |
| Responses to the Environment | electrical troubles, furniture damages, items falling over, answering calls |
| Interactions with Partner | call for help, share, make way, collaborate, correct |

providing synchronized SMPL-based human motion, multi-view RGB-D videos, detailed 3D scene information, and natural language descriptions of human intentions. For each sequence, the 3D scene information includes a colored, instance-labeled point cloud of the entire scene, along with a 3D mesh model in which the scanned meshes of the objects are correctly positioned within the scene. The motion data captured at 240 fps consists of 17.5M frames in total, with a reliability flag provided for each frame to indicate data trustworthiness (e.g., for moments when a subject is temporarily outside the capture volume). The RGB-D video data comprises 436K frames recorded at 30 fps from four perspectives. Each sequence has an average duration of 169 seconds. The dataset includes a total of 1,388 textual annotations for intentions. Figure 2 showcases the modalities available in our dataset.

## 3.2 Data Collection Design

To capture motions that are both realistic and grounded in purpose, our data collection was built upon three key design pillars.

### 3.2.1 Human Intention Design

In our dataset, human intentions are classified into four main categories: long-term activities, short-term needs, responses to the environment, and interactions with partner. Table 2 provides illustrative examples for each category.

**Long-term Activities.** We defined several common indoor activities, such as leisure, studying, cleaning, and exercising. These activities typically span longer periods and form the primary context for each motion sequence in the dataset. Further examples are detailed in Table 5.

**Short-term Needs.** These include intentions like drinking water, eating, or resting. Such needs are not always directly related to the ongoing long-term activity and can arise at any point, causing shifts in the subsequent human motion.

**Responses to the Environment.** Individuals continuously perceive and react to their surroundings. This type of response, such as answering an unexpected phone call or reacting to a falling object, is treated as a latent intention that influences motion.

**Interactions with Partner.** As our scenes always include two individuals, interaction is a key element. This includes intentions to collaborate, seek help from the partner, or respond to actions initiated by them.

### 3.2.2 Designing for Behavioral Diversity

A core challenge in modeling human behavior is its inherent ambiguity: a single high-level goal can lead to a wide variety of plausible physical actions, framing the task as a one-to-many mapping problem. While an intention provides a strong constraint, it does not completely eliminate this ambiguity. For instance, the intention "get a drink" could result in walking to the kitchen to use the tap, opening the refrigerator for a bottle of water, or asking a partner to pass a drink. Each outcome involves a distinct motion sequence. To facilitate the study of this complex distribution, we therefore captured this behavioral diversity by having a range of participants perform similar tasks multiple times under varied conditions.

### 3.2.3 Scene Design

Our scenes are realistic home environments constructed with common household furniture and items, designed to facilitate natural interactions based on the subjects' intentions. The scenes have the following key characteristics:

**Dynamic Nature.** Beyond interactions from the subjects, we incorporated dynamic elements common in a home environment, such as an electric kettle boiling water or a phone ringing, to create a more realistic and unpredictable setting.

**Partial Visibility.** We distinguish between visible and invisible parts of the scene. Areas perceived by the sensors are considered visible. However, other rooms or the contents of containers like cabinets and refrigerators are invisible, reflecting real-world scenarios where complete information is unavailable.

**Human Perception.** Subjects perceive the visible parts of the scene, including dynamic changes, and their actions are driven by their intentions in response to what they perceive. Their knowledge of invisible parts may be incomplete or unreliable, and it is updated through interaction. For example, a person might discover a magazine under the sofa while cleaning or mistakenly search for an item in the wrong cabinet.

## 3.3 Data Acquisition and Annotation

Our process for creating the dataset involved careful planning of scenarios, a high-fidelity recording setup, and a detailed annotation pipeline.

### 3.3.1 Recording Environment and Procedure

The dataset was collected in a motion capture studio arranged to simulate a home environment. The setup includes a synchronized system of 12 OptiTrack Prime X13 optical motion capture cameras and 4 Kinect sensors. After calibrating all sensors and participants' skeletons, we guided their actions with textual instructions according to pre-designed scenarios that combined several long-term activities and short-term needs. To enhance data diversity, the specifics of the activities and scene layouts were frequently changed. Subjects wore everyday clothing fitted with small, unobtrusive optical markers, allowing us to capture high-precision kinematics while maintaining a natural visual appearance in the RGB-D data. Following the approach of [Zheng et al., 2022], we captured high-quality 3D meshes by scanning the environment using the Scaniverse[1] app on an iPhone 15 Pro equipped with a LiDAR sensor. Further details of the recording environment are provided in Section A.

---
[1] https://scaniverse.com

### 3.3.2 Processing and Annotation

The collected raw data was processed and annotated to generate the final dataset modalities:

**Human Motion Data.** The captured marker data was processed using Motive 3.0.3 to compute 3D joint positions and orientations. We then fit this data to the SMPL model to obtain full-body pose. To handle situations where motion capture might be unreliable—for instance, when subjects are temporarily outside the capture volume to retrieve an item from another room or when a guest enters the scene mid-sequence—we provide a reliability flag for each motion frame indicating whether the data is available and trustworthy.

**Intention Annotations.** After collection, we reviewed the synchronized video and motion data to manually annotate the precise start frame for each textual goal, creating a detailed, timestamped log of each participant's intentions.

**3D Scene Data.** For each sequence, we provide two forms of 3D data: (1) a comprehensive colored 3D point cloud of the entire scene, where each point is annotated with an object instance label, and (2) a 3D mesh model of the scene in which the scanned object meshes are accurately placed. The dynamic aspects of the scene are captured by the multi-view RGB-D videos.

## 3.4 Intended Uses

I2M opens up new possibilities for studying human behavior and motion. Its long-term, intention-driven sequences support research in areas such as motion generation, motion prediction, and action planning, where agents must reason about high-level goals in complex environments. The inclusion of text and scene information enables multimodal approaches with potential applications in real-world settings. Additionally, the dataset offers rich data on multi-person interactions, facilitating studies in social robotics and human-human collaboration. Beyond specific prediction tasks, I2M provides a foundation for exploring how human intentions can be inferred from observed motions—a crucial step toward developing more intelligent and empathetic AI systems.

# 4 Experiment

## 4.1 Problem Setting

We tackle the task of long-term human motion prediction in dynamic home environments using the I2M dataset. This task aims to predict a person's future motion in contexts with rich scene information and human-human interactions. Specifically, we formulate the task as predicting the next 5 seconds (150 frames at 30 fps) of motion, conditioned on the past 3 seconds (90 frames). Motions are represented as 3D joint positions based on the SMPL joint format. Data sequences were generated using a 60-frame stride sliding window. This process resulted in 25,000 training data samples and 606 test data samples.

## 4.2 Prediction Model and Evaluation Metrics

**Prediction Model.** We adapt the AffordMotion model [Wang et al., 2024], a diffusion-based motion generation framework. The prediction task is formulated as a conditional inpainting problem. The model is trained to denoise a full 240-frame sequence (90 past, 150 future). We start with a random noise sequence and replace the first 90 frames with the ground-truth past motion. The model then denoises the whole frames to generate the future motion prediction, conditioned on the provided past motion and other semantic inputs.

**Conditioning Modalities.** In addition to the standard scene point cloud input, we investigate various conditioning modalities, each embedded into a 512-dimensional vector. These include: `Intention Text`, natural language descriptions of human intentions encoded via CLIP's text encoder [Radford et al., 2021]; `RGB Images`, multi-view frames encoded by a ResNet-50 [He et al., 2016]; `Trajectory`, the past root joint trajectory encoded by an MLP; and `MotionCLIP`, semantic embeddings of past motion from the MotionCLIP model [Tevet et al., 2022].

Table 3: **Quantitative evaluation results under different input conditions.** We use colors to denote the first and second places respectively.

| Condition | ADE@1 ↓ | FDE@1 ↓ | MPJPE@1 ↓ | ADE@20 ↓ | FDE@20 ↓ | MPJPE@20 ↓ |
|---|---|---|---|---|---|---|
| Self Intention Text | $1.573 \pm 0.002$ | $1.584 \pm 0.010$ | $1.606 \pm 0.002$ | $0.867 \pm 0.002$ | $0.785 \pm 0.004$ | $0.942 \pm 0.002$ |
| RGB Images | $1.651 \pm 0.002$ | $1.671 \pm 0.005$ | $1.709 \pm 0.003$ | $0.659 \pm 0.001$ | $0.634 \pm 0.004$ | $0.776 \pm 0.001$ |
| Self Trajectory | $1.175 \pm 0.002$ | $1.255 \pm 0.008$ | $1.264 \pm 0.001$ | $0.583 \pm 0.002$ | $0.556 \pm 0.003$ | $0.732 \pm 0.001$ |
| Pair Trajectory | $1.013 \pm 0.002$ | $1.099 \pm 0.004$ | $1.155 \pm 0.002$ | $0.490 \pm 0.002$ | $0.479 \pm 0.002$ | $0.678 \pm 0.002$ |
| Self MotionCLIP | $1.434 \pm 0.002$ | $1.470 \pm 0.006$ | $1.474 \pm 0.002$ | $0.831 \pm 0.001$ | $0.787 \pm 0.002$ | $0.915 \pm 0.001$ |
| Pair MotionCLIP | $1.355 \pm 0.002$ | $1.406 \pm 0.004$ | $1.407 \pm 0.002$ | $0.653 \pm 0.002$ | $0.613 \pm 0.005$ | $0.756 \pm 0.002$ |
| Self Multimodal | $1.336 \pm 0.002$ | $1.423 \pm 0.005$ | $1.390 \pm 0.002$ | $0.749 \pm 0.002$ | $0.731 \pm 0.004$ | $0.840 \pm 0.002$ |
| Pair Multimodal | $1.220 \pm 0.002$ | $1.260 \pm 0.004$ | $1.289 \pm 0.002$ | $0.657 \pm 0.001$ | $0.621 \pm 0.003$ | $0.761 \pm 0.001$ |

| Condition | MMADE@20 ↓ | MMFDE@20 ↓ | MCE@20 ↓ | APD@20 ↑ | Non-coll@1 ↑ | Contact@1 ↑ |
|---|---|---|---|---|---|---|
| Self Intention Text | $1.140 \pm 0.001$ | $1.087 \pm 0.002$ | $1.499 \pm 0.003$ | $5.530 \pm 0.005$ | $0.998 \pm 0.000$ | $0.672 \pm 0.005$ |
| RGB Images | $1.159 \pm 0.000$ | $1.152 \pm 0.001$ | $1.321 \pm 0.003$ | $7.461 \pm 0.002$ | $0.998 \pm 0.000$ | $0.730 \pm 0.002$ |
| Self Trajectory | $0.820 \pm 0.000$ | $0.828 \pm 0.001$ | $1.125 \pm 0.002$ | $4.666 \pm 0.001$ | $0.997 \pm 0.000$ | $0.858 \pm 0.002$ |
| Pair Trajectory | $0.709 \pm 0.000$ | $0.729 \pm 0.000$ | $1.191 \pm 0.003$ | $4.281 \pm 0.002$ | $0.997 \pm 0.000$ | $0.848 \pm 0.003$ |
| Self MotionCLIP | $1.045 \pm 0.001$ | $1.039 \pm 0.001$ | $1.218 \pm 0.002$ | $4.546 \pm 0.002$ | $0.998 \pm 0.000$ | $0.806 \pm 0.001$ |
| Pair MotionCLIP | $0.985 \pm 0.000$ | $0.973 \pm 0.001$ | $1.112 \pm 0.003$ | $5.310 \pm 0.004$ | $0.997 \pm 0.000$ | $0.849 \pm 0.002$ |
| Self Multimodal | $0.970 \pm 0.000$ | $0.998 \pm 0.001$ | $1.181 \pm 0.004$ | $4.350 \pm 0.003$ | $0.997 \pm 0.000$ | $0.822 \pm 0.003$ |
| Pair Multimodal | $0.853 \pm 0.000$ | $0.835 \pm 0.001$ | $1.149 \pm 0.003$ | $3.992 \pm 0.001$ | $0.998 \pm 0.000$ | $0.812 \pm 0.002$ |

**Experimental Setup.** We design eight distinct conditioning configurations: `Self Intention Text`, `RGB Images`, `Self Trajectory`, `Pair Trajectory`, `Self MotionCLIP`, `Pair MotionCLIP`, `Self Multimodal`, and `Pair Multimodal`. "Self" configurations use information pertaining only to the person whose motion is being predicted, while "Pair" configurations additionally incorporate information from the interaction partner. The `Self Multimodal` and `Pair Multimodal` configurations combine all respective "Self" and "Pair" modalities to assess their synergistic effect. For each configuration, we fine-tune the pretrained AffordMotion model for 20,000 steps. We repeat each experiment 5 times with different random seeds and report the mean and standard deviation of the metrics. Further details are provided in Section B.1

**Evaluation Metrics.** We evaluate the generated motions using a comprehensive set of metrics. For prediction accuracy, we use the standard Average Displacement Error (**ADE**), Final Displacement Error (**FDE**), and Mean Per Joint Position Error (**MPJPE**). We report these for a single prediction and the best among K=20 samples. To evaluate the model's ability to capture the multimodal nature of human behavior, where multiple future paths are plausible, we use Multimodal ADE (**MMADE**) and Multimodal FDE (**MMFDE**) [Yuan and Kitani, 2020]. These metrics assess how well the distribution of generated samples covers the multiple potential ground truth futures. Furthermore, we introduce MotionCLIP Error (**MCE**) to measure perceptual similarity based on the L2 distance between the MotionCLIP embeddings of the predicted and ground truth motions. To measure the diversity of the generated samples, we use Average Pairwise Distance (**APD**). For physical plausibility, we adopt Non-collision (**Non-coll**) and **Contact** from [Wang et al., 2024] to evaluate realism within the scene context. Detailed definitions for these metrics are provided in Section B.2.

## 4.3 Quantitative Results

Table 3 shows the quantitative results. Trajectory-only models showed superior performance across all metrics, likely due to their clear spatial-temporal structure. The performance boost from `Pair Trajectory` over `Self Trajectory` indicates that even simple conditioning enables the model to capture human-human interactions.

In contrast, conditioning on semantically rich inputs such as RGB images and intention text led to greater motion diversity, as measured by APD. However, this increase in APD does not necessarily correspond to semantically meaningful diversity. In particular, the limited improvement observed in multimodal metrics such as MMADE and MMFDE suggests that the generated samples do not consistently align with the distribution of plausible future motions. Instead, the higher APD likely reflects increased model uncertainty in grounding semantic cues into motion, resulting in more divergent but less well-grounded predictions.

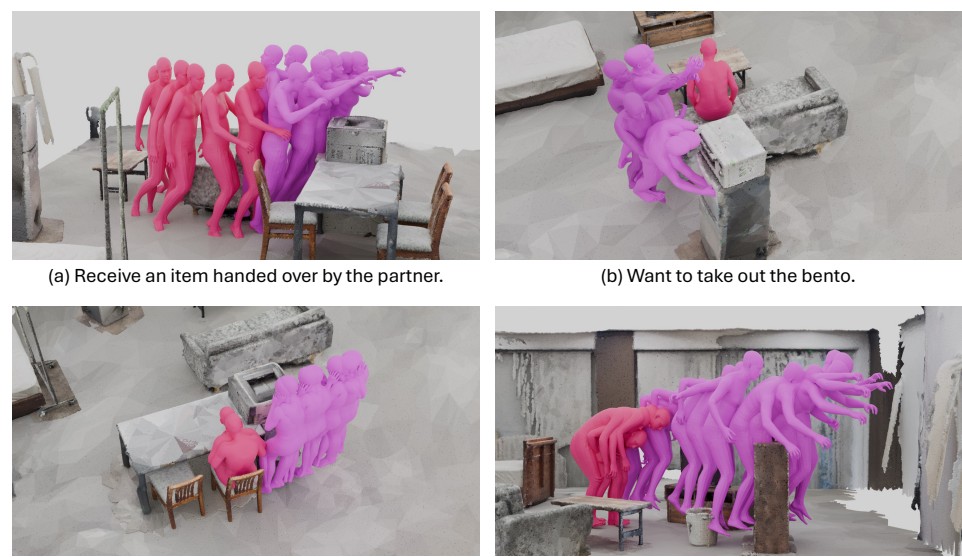

| (a) Receive an item handed over by the partner. | (b) Want to take out the bento. |
| (c) Today's work is done. | (d) Clean the room. |

Figure 3: **Qualitative results.** Qualitative examples from our `Pair Multimodal` model illustrate both successful outcomes (a–c) and a common failure mode (d). The input motion is shown in dark pink, and the predicted future motion is in light purple. While the model can generate motions consistent with the given intention, it sometimes produces physically implausible results (e.g., object penetration), highlighting key challenges for future research.

### 4.4 Qualitative Results

To provide a qualitative assessment of our baseline model's performance, Figure 3 presents several examples under the `Pair Multimodal` condition. We first examine successful cases where the model generates motions semantically consistent with the given intentions. For instance, given the intention "Receive an item handed over by the partner," the model correctly predicts a motion of walking toward the partner and extending an arm to receive the item (Figure 3-a). For "Want to take out the bento," it generates a plausible sequence of the human standing up from a sofa and walking to the refrigerator (Figure 3-b). For "Today's work is done," the model produces a natural sequence of the human standing up from their desk and moving away (Figure 3-c).

However, the results also highlight common failure modes. As shown in Figure 3-d, the model can generate physically implausible motions. In this example, the generated motion for "Clean the room" shows the human's body passing through a bookshelf while their feet float above the ground. This example illustrates the remaining challenges in grounding high-level intentions into physically realistic and scene-aware motion, underscoring the research opportunities enabled by our dataset.

### 4.5 Discussion

A key finding is that the Pair Multimodal model, despite having access to all available data modalities, exhibited weaker performance on quantitative metrics than models conditioned solely on trajectory information. This suggests that incorporating semantically rich inputs—such as intention text, RGB images, and MotionCLIP embeddings—does not directly translate into reduced positional error under current modeling choices. Importantly, this behavior does not indicate a limitation of the dataset itself, but rather exposes a gap between rich semantic observations and existing motion generation architectures. Pre-trained encoders such as CLIP and ResNet, while effective for general semantic representation, may not yield embeddings that are readily compatible with temporally coherent and physically plausible motion generation in complex scene contexts. Taken together, these observations highlight a fundamental architectural challenge: current models struggle to translate high-level semantic goals into the low-level kinematics required for motion generation.

Furthermore, a critical limitation in the field is the absence of metrics designed to evaluate whether a generated motion fulfills the underlying human intention. Current geometric metrics like ADE and FDE measure positional accuracy but fail to capture this higher-level concept of success. For

example, consider the intention "take a book from the shelf." A predicted motion might bring the person close to the shelf, resulting in a low geometric error. However, if the person never actually touches the book, the motion has failed to achieve the core intention. Even our semantic metric, MotionCLIP Error (MCE), only assesses consistency within a learned feature space and falls short of evaluating whether the intention was truly fulfilled.

Based on these findings, future work should address two key areas. The first is the development of new architectures capable of aligning high-level semantic intention with low-level motion trajectories over time. The second is the establishment of a new evaluation framework that moves beyond geometric error to directly measure how successfully a generated motion achieves its stated intention.

The I2M dataset provides the resources to tackle these fundamental challenges. It provides rich, multimodal data grounded in explicit human intention. This creates a foundation for developing models that can reason about high-level goals, alongside evaluation frameworks capable of measuring their successful execution.

## 5 Limitations and Future Directions

Our data collection methodology presents clear directions for future extensions. While our focus was on full-body kinematics, the current dataset does not include fine-grained hand and finger articulations, which are critical for detailed interaction analysis. Furthermore, our setup lacks egocentric vision or gaze tracking, modalities that offer a direct window into a human's focus of attention and intent, and our use of textual instructions precluded the capture of natural conversational audio.

To further enhance value and generalizability, future efforts should aim to capture a broader spectrum of behavioral diversity. This would involve recruiting a wider range of participants, considering factors like age and personality traits which can lead to significant variations in motion patterns. Expanding the scope beyond the current two-person, indoor scenarios to include outdoor environments and larger group interactions would also be a valuable direction, enabling the study of more complex social and environmental dynamics.

Beyond the data itself, our work highlights the urgent need for new evaluation paradigms within intention-aware motion modeling. Traditional geometric metrics such as ADE and FDE are insufficient for this task, as they primarily measure spatial similarity and cannot confirm if a semantic goal was successfully achieved. A critical avenue for future research is therefore the creation of robust "intention-aware" metrics that move beyond positional accuracy to measure true goal fulfillment. This might involve defining task-specific success criteria or leveraging multimodal large language models to assess the semantic plausibility of generated motions.

## 6 Conclusion

We introduced *Intend to Move (I2M)*, a large-scale multimodal dataset designed to shift human motion research from describing "what" humans do to understanding "why" they move. By capturing long-duration, multi-person motion sequences grounded in explicit intentions and rich, dynamic scenes, I2M establishes a foundation for studying socially aware, intention-driven behavior in realistic contexts. Our experiments reveal a persistent gap between the high-level semantic reasoning required to interpret human intention and the capabilities of current motion generation models. This gap highlights that advancing the field demands more than scaling data—it requires rethinking the architectures and objectives of motion modeling itself. We thus present I2M not merely as a dataset, but as a catalyst for developing models that can ground abstract goals into physical action, and for designing evaluation paradigms that assess true semantic goal fulfillment. Ultimately, I2M aims to pave the way toward the next generation of embodied AI systems—ones that can genuinely understand, predict, and collaborate with humans in the complex, dynamic environments we share.

## Acknowledgements

This work was partially supported by JST Moonshot R&D Grant Number JPMJPS2011, CREST Grant Number JPMJCR2015 and Basic Research Grant (Super AI) of Institute for AI and Beyond of the University of Tokyo.

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

# A  Dataset Details

We constructed a multimodal capture environment within a 7m x 7m space, as illustrated in Figure 6. This environment was equipped with a system of 12 OptiTrack Prime X13 optical motion capture cameras (processing the captured marker data with Motive 3.0.3) for high-fidelity motion tracking. Simultaneously, four Kinect cameras were positioned at the corners of the space to record synchronized multi-view RGB-D video.

Our data collection process was meticulously designed to address the challenges outlined in Section 2, specifically the need to unite high motion accuracy with visual realism. The key to our approach lies in bridging the visual domain gap inherent in many optical capture datasets: instead of using specialized mocap suits, subjects wore everyday clothing onto which we affixed small, visually unobtrusive optical markers, as shown in Figure 4. This method allowed us to capture both high-precision kinematic data and visually authentic footage of subjects in natural attire.

To ensure the captured interactions were grounded in a realistic context, the scene was furnished with common household items and furniture, creating a dynamic and interactive home environment. A comprehensive list of these items is provided in Table 4. Samples of the resulting synchronized motion and RGB data can be seen in Figure 7.

# B  Experimental Details

## B.1  Implementation Details

All models were trained on a single NVIDIA A100 GPU (80GB) using a batch size of 256 and a learning rate of $1 \times 10^{-4}$. For motion feature extraction, we utilized the MotionCLIP [Tevet et al., 2022] model with the officially released pretrained weights. For RGB image feature extraction, we employed a ResNet-50 [He et al., 2016] model sourced from torchvision version 0.19.1 [maintainers and contributors, 2016].

## B.2  Metrics Definition

We use the following metrics to measure the performance. The "@K" notation indicates that the metric is evaluated over K generated samples. For instance, ADE@1 considers only a single prediction, while ADE@20 considers the best among 20 samples.

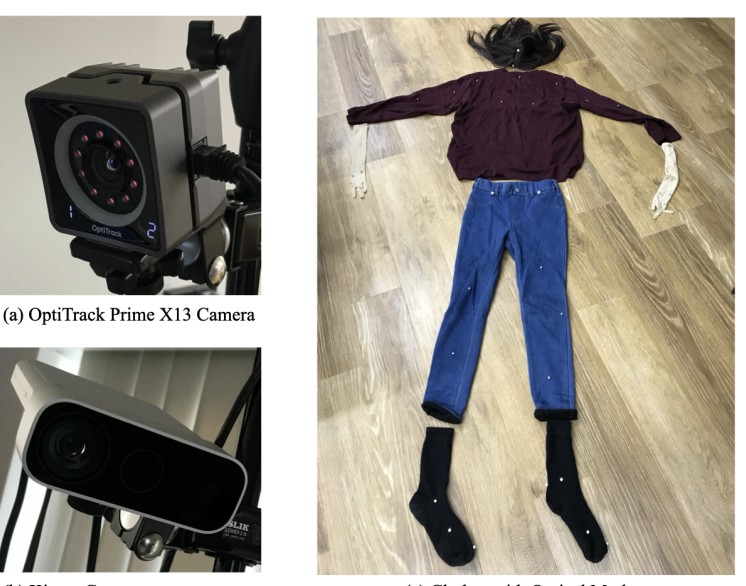

(a) OptiTrack Prime X13 Camera

(b) Kinect Camera

(c) Clothes with Optical Markers

Figure 4: **Devices and marker-fitted clothing used for data collection.**

**(1) Average Displacement Error (ADE):**  The average $L_2$ distance over all $T$ time steps between the ground truth motion $\hat{\mathbf{p}}$ and the closest predicted sample $\mathbf{p}_k$ among $K$ samples. It is computed as:

$$\text{ADE@K} = \min_{k \in \{1,\ldots,K\}} \left( \frac{1}{T} \sum_{t=1}^{T} \|\mathbf{p}_{k,t} - \hat{\mathbf{p}}_t\|_2 \right)$$

where $\mathbf{p}_{k,t}$ is the predicted pose at timestep $t$ for sample $k$, and $\hat{\mathbf{p}}_t$ is the ground truth pose at timestep $t$.

**(2) Final Displacement Error (FDE):**  The $L_2$ distance between the final ground truth pose $\hat{\mathbf{p}}_T$ and the final pose of the closest sample $\mathbf{p}_{k,T}$ among $K$ samples:

$$\text{FDE@K} = \min_{k \in \{1,\ldots,K\}} \|\mathbf{p}_{k,T} - \hat{\mathbf{p}}_T\|_2$$

**(3) Mean Per Joint Position Error (MPJPE):**  The average $L_2$ error across all $J$ joints and $T$ timesteps for the closest sample among $K$ predictions:

$$\text{MPJPE@K} = \min_{k \in \{1,\ldots,K\}} \left( \frac{1}{T \cdot J} \sum_{t=1}^{T} \sum_{j=1}^{J} \|\mathbf{p}_{k,t,j} - \hat{\mathbf{p}}_{t,j}\|_2 \right)$$

where $\mathbf{p}_{k,t,j}$ is the position of joint $j$ at timestep $t$ for sample $k$.

**(4) Average Pairwise Distance (APD):**  The average $L_2$ distance between all pairs of $K$ motion samples, measuring diversity:

$$\text{APD@K} = \frac{1}{K(K-1)} \sum_{i=1}^{K} \sum_{j \neq i, j=1}^{K} \left( \frac{1}{T} \sum_{t=1}^{T} \|\mathbf{p}_{i,t} - \mathbf{p}_{j,t}\|_2 \right)$$

A higher APD indicates greater diversity among the generated samples.

**(5) Multimodal ADE (MMADE) and Multimodal FDE (MMFDE):**  These are multimodal versions of ADE and FDE, respectively [Yuan and Kitani, 2020]. They are designed to evaluate a method's ability to produce diverse predictions that cover multiple potential ground truth future motions, particularly when the future is inherently ambiguous.

Standard metrics like ADE@K evaluate $K$ generated samples against a *single* ground truth sequence. This penalizes diverse but plausible predictions that deviate from that specific ground truth. To address this, MMADE and MMFDE evaluate the generated distribution against a *set of plausible ground truth futures*.

First, for a given input motion, we construct a set of plausible ground truth future motions, $\mathcal{G}$. This set is formed by collecting all motions from the test set whose starting pose is within a predefined distance threshold to the final pose of the given input motion. Let $\mathcal{P} = \{\mathbf{p}_1, \ldots, \mathbf{p}_K\}$ be the set of $K$ predicted motions generated by the model.

**MMADE@K** is calculated by first finding the ADE between each predicted motion $\mathbf{p}_k \in \mathcal{P}$ and its closest motion in the plausible set $\mathcal{G}$. These "best-match" ADEs are then averaged over all $K$ predictions. This rewards models that generate a diverse set of samples, where each sample is close to at least one of the plausible future outcomes. The formula is:

$$\text{MMADE@K} = \frac{1}{K} \sum_{k=1}^{K} \left( \min_{\hat{\mathbf{g}} \in \mathcal{G}} \left( \frac{1}{T} \sum_{t=1}^{T} \|\mathbf{p}_{k,t} - \hat{\mathbf{g}}_t\|_2 \right) \right)$$

where $\mathbf{p}_{k,t}$ is the pose of the $k$-th prediction at time $t$, and $\hat{\mathbf{g}}_t$ is the pose of a ground truth motion from the plausible set $\mathcal{G}$ at time $t$.

**MMFDE@K** follows the same logic, but for the final displacement error. For each of the $K$ predicted motions, we find the FDE to its closest counterpart in the plausible set $\mathcal{G}$ and then average these values.

$$\text{MMFDE@K} = \frac{1}{K} \sum_{k=1}^{K} \left( \min_{\hat{\mathbf{g}} \in \mathcal{G}} \|\mathbf{p}_{k,T} - \hat{\mathbf{g}}_T\|_2 \right)$$

where $\mathbf{p}_{k,T}$ and $\hat{\mathbf{g}}_T$ represent the final poses of the predicted and ground truth motions, respectively.

Request for Your Cooperation in Research on Context-Aware Human Motion Prediction from Video Using a Real-World Multimodal Dataset

This document explains the research project titled "Context-Aware Human Motion Prediction from Video Using a Real-World Multimodal Dataset." Please be assured that you will not be disadvantaged in any way if you choose not to participate in this study. If you have any questions or if anything is unclear, please do not hesitate to ask the research staff.

1. Overview of This Research

- Research Topic: Context-Aware Human Motion Prediction from Video Using a Real-World Multimodal Dataset.

- Research Purpose: The purpose of this research is to develop a method for predicting human motion that considers collaborative actions between people and the surrounding environment. To achieve this, we will create a dataset of actual human movements and use it to build and evaluate our proposed method.

- Research Method: You will be asked to wear clothing fitted with markers for the OptiTrack motion capture system and perform actions as instructed in writing. Your motion data during these actions will be measured and recorded using these markers. Simultaneously, your movements will be recorded by cameras.

  You will participate in the experiment for a duration previously communicated to you. Additionally, participants who have been informed in advance will consume a meal as previously instructed.

  Individuals who are not between the ages of 20 and 65 (inclusive) cannot participate in this experiment. Furthermore, this study is intended only for healthy individuals; therefore, those with limb deficiencies or difficulties with physical movement are not eligible to participate.

  The OptiTrack motion capture system will be operated by researchers who are thoroughly familiar with it, and safety will be a constant consideration. In the unlikely event that you sustain an injury or feel unwell, the experiment will be paused. If your condition does not improve after a break, the experiment will be terminated.

2. Voluntary Participation and Freedom to Withdraw

Your participation in this research is entirely voluntary. If you agree to participate but later decide to withdraw your consent, please sign a withdrawal form and submit it to the contact address provided below. Please be assured that refusing to participate or withdrawing from the study will not result in any disadvantage to you.

If you withdraw your consent, any samples and information you provided will be disposed of and will not be used for further research. However, please understand that in the following cases, it may not be possible to dispose of your samples and information even if you withdraw consent:

- If individuals cannot be identified from the provided samples and information.
- If data analysis has already been conducted, and it is not possible to separate and dispose of your specific samples and information.
- If the data has already been published in papers, preprints, data sharing servers, or conference presentations.

3. Protection of Personal Information

We will make every effort and take the utmost care to protect your personal information and respect your privacy to ensure that no disadvantage occurs to you. Personal identifiers such as your name will be removed from your samples and information, and a new code will be assigned for research purposes. Consent forms containing your name will be stored securely in a locked document storage unit within a facility managed by the principal investigator, under strict and responsible management.

4. Publication and Disclosure of Research Results

The results of this research, with names removed, including video data (which may include faces) and motion data (including walking), will be published in academic conferences, scholarly journals, and on the project's webpage, etc.

5. Benefits and Disadvantages for Research Participants

It is currently considered unlikely that this research will provide immediate beneficial information to you or society. However, the findings of this study are expected to become important foundational results that will contribute to the future development of machine learning research.

On the other hand, foreseeable disadvantages include feeling unwell during the experiment, falls, or collisions with equipment. However, we will take measures such as monitoring your physical condition during the experiment, providing breaks if you feel unwell, scheduling regular breaks, discontinuing the experiment if necessary, and implementing accident prevention measures beforehand.

6. Policy on Handling of Samples and Information

Samples and information provided by you will be used for research and analysis after your name has been removed. With your consent, these valuable samples and information may be stored beyond the initial retention period for future research. Your samples and information, with your name removed, may be used by our institution, provided to domestic or international institutions, and are planned to be publicly released on the project's webpage.

7. Your Financial Burden

You will not be responsible for any costs associated with this research. Compensation for your participation in this study will be provided as previously communicated.

Figure 5: **Instructions presented to the subjects for the data collection.**

## C  Ethics Statement

The human subject study in this paper has been reviewed and approved by Research Ethics Committee of the University of Tokyo. In the study, we informed the subjects in advance that their facial and motion data would be publicly disclosed for research purposes. With their consent, we had them sign both a consent form and a consent withdrawal form. The instructions presented to the subjects in the experiment are shown in Figure 5.

## D  Potential Negative Impact

The dataset we provide is intended for human motion modeling, and it could lead to surveillance of individuals or privacy violations if misused with malicious intent. Furthermore, because we provide a set of human motion and intentions, it may be harmful if a model is trained to infer intentions from behavior. To reduce such risks, we implement the access control and strictly review those who are eligible for access. We will also suspend the access permission at any time if we discover someone using it improperly.

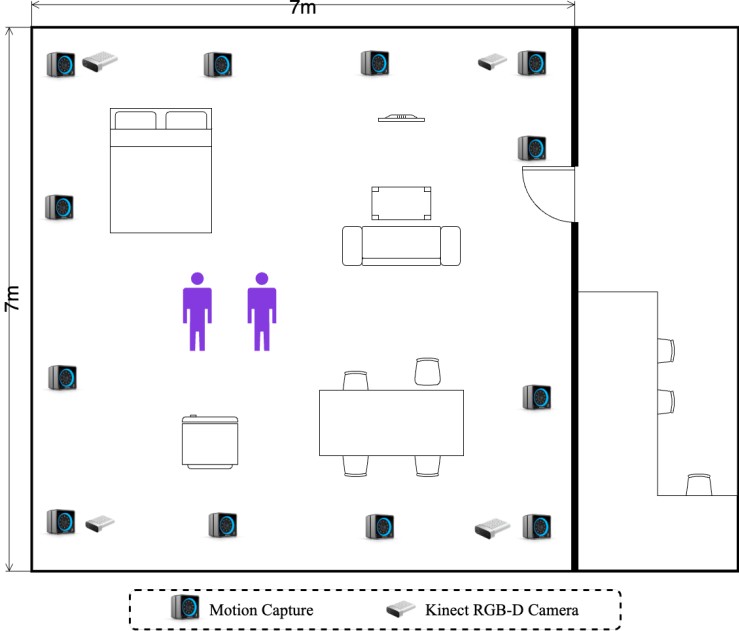

Figure 6: **Layout of the data collection environment.** The 7m × 7m capture area (left) is surrounded by sensors. The workstation (right) is located outside the sensors' field of view, allowing researchers to monitor the process and instruct subjects. The furniture positions shown are illustrative, as the scene layout was frequently changed.

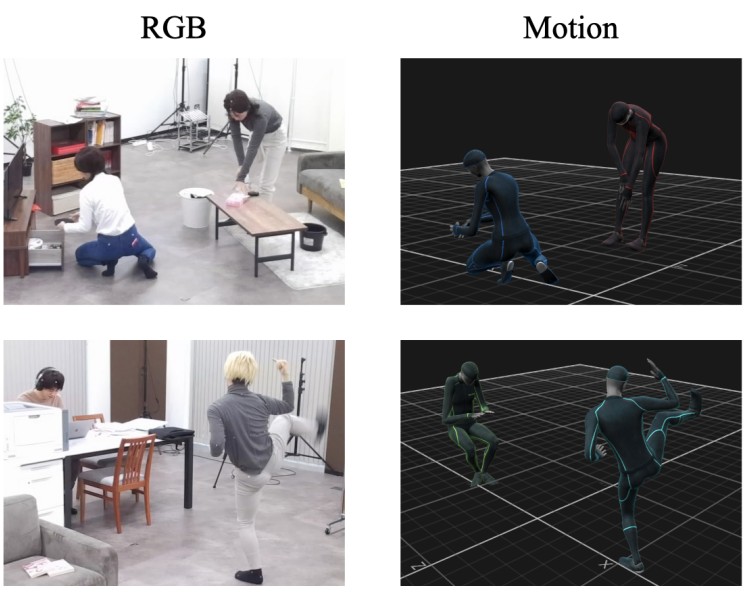

Figure 7: **Synchronized RGB images and their corresponding motion data.**

Table 4: **Items appearing in the I2M dataset.**

| Object categories | Items |
|---|---|
| Furniture | sofa, table, chair, bed, book shelf, cabinet |
| Appliances | television, TV remote, refrigerator, kettle, printer, monitor, lamp, washing machine |
| Housekeeping-related items | vacuum cleaner, trash bin, garbage bag, towel, mop, broom, laundry detergent, clothes, clothes hanger, pillow, pillowcase, newspaper, tissue box |
| Study / office items | book, laptop, paper, whiteboard, whiteboard marker, headphones |
| Food-related items | teapot, teacup, paper cup, tray, dish, beverage bottle, beverage carton, fruit, beverage can, snack, bento, paper-wrapped food, ice cream, cup noodles, straw, tea leaves in a box, chopsticks, spoon, coffee maker, capsule for coffee maker |
| Personal items | mobile phone, watch, wallet, handbag |
| Leisure / sports items | dumbbell, yoga mat, comics, chess, Jenga |

Table 5: **Examples of intentions related to long-term activities.**

| Long-term Categories | Descriptions |
|---|---|
| **Leisure** | Want to watch TV |
| | Have nothing to do |
| | Want to sleep |
| | Want to read comics for fun |
| | Want to enjoy music |
| **Studying** | Is studying |
| | Have to study afterward |
| | There are unclear points in the knowledge from the book |
| | Want to use reference books in study |
| **Online Meeting** | Is attending an online meeting |
| | Have to attend an online meeting afterward |
| | Want to prepare materials for an online meeting |
| | Have to give a presentation in an online meeting |
| **Entertaining Friends** | A friend is about to visit |
| | A friend is visiting |
| | Serving tea to a friend |
| | Playing games with a friend |
| | Seeing a friend off |
| | Giving a gift to a friend |
| **Cleaning** | Want to organize clothes |
| | Want to wash clothes |
| | Want to tidy up clutter |
| | Want to clean the room |
| **Exercising** | Want to do warm-up exercises |
| | Want to use dumbbells for strength training |
| | Want to do abdominal exercises |
| | Want to practice karate |

