# OpenReview forum: "Intend to Move: A Multimodal Dataset for Intention-Aware Human Motion Understanding"
_NeurIPS.cc/2025/Datasets_and_Benchmarks_Track — NeurIPS 2025 Datasets and Benchmarks Track poster_

### Official Review · Reviewer_nZdu · 2025-06-02

**Rating:** 4
**Confidence:** 4

**Summary:**

This paper proposes the Intend to Move (I2M) dataset, which aims to bridge the gap of intention-driven human motion prediction in existing research. The proposed I2M dataset comprises 215 temporally continuous sequences totaling 10.1 hours of motion data, annotated with SMPL pose, multi-view RGB-D video, scene point clouds, and 38 high-level textual goals. Evaluation results using an adapted AffordMotion model show that the task remains challenging on existing motion prediction metrics.

**Dataset Code Accessibility:**

No

**Dataset Code Comments:**

The dataset is not quite accessible as it does not include a document explaining its structure and the meaning of each file, or how to use the proposed dataset.

The toolbox folder is empty and one cannot reuse the tools.

**Ethical Considerations:**

No, there are no or only very minor ethics concerns

**Final Justification:**

Thank the authors for their reply and the rebuttal partially resolves my concerns. However, to better facilitate the community, the fundamental limitations of the dataset should be clearly articulated in the revised version.

Therefore, I am happy to raise my rating to borderline accept.

**Limitations Weaknesses:**

The key limitation of this dataset is that it is still preliminary and far from sufficient to resolve the challenge. Specifically, as the authors mentioned in Lines 125 -127 "... the theoretical feasibility of supporting the task of motion prediction depends on whether the task setting provides sufficient semantic information required for prediction", having intention information does not reduce much of the ambiguity in motion prediction (e.g., with the same intention, different people will produce different motion sequences and even the same people will produce different motion sequences at different timestamps/locations). Therefore, the information proposed in the dataset is insufficient to resolve the challenge and may not be very helpful to the community.

Thanks to the remaining motion ambiguity, a valid metric for this task is to evaluate whether the resulting motion sequences are reasonable for the given intention rather than comparing them to the "ground truth" using error metrics (e.g., ADE, FDE). Therefore, the evaluation/benchmark part is less valid as well.

The dataset is not quite accessible as it does not include a document explaining its structure and the meaning of each file, or how to use the proposed dataset. The toolbox folder is empty and one cannot reuse the tools.

**Strengths Contributions:**

The paper is well-written and easy to follow.

The work proposes an interesting task and the dataset proposed is clearly different from previous ones in providing the intention text/annotations (Table 1 in the paper).

The data is collected from complex real-world scenarios with two people interacting with each other, which requires a lot of work.

---

> ### Author Rebuttal · Authors · 2025-07-29
>
> We thank the reviewer for their time and their detailed, constructive feedback. We are encouraged that the reviewer recognized that our work **"proposes an interesting task,"** with a dataset that is **"clearly different from previous ones,"** and was **"collected from complex real-world scenarios"** which **"requires a lot of work."**
>
> The reviewer has raised several critical points that have given us an opportunity to substantially improve our work. We address these fundamental concerns below.
>
> &nbsp;
> ### **1. On the Ambiguity of Motion**
>
> We thank the reviewer for this insightful point. We completely agree that motion prediction is an inherently ambiguous (one-to-many) task. In fact, **a primary contribution of our work is to formalize this challenging one-to-many prediction problem and provide the first large-scale benchmark for studying it.**
>
> We hypothesize that a high-level intention, while not eliminating ambiguity, serves as a powerful constraint that **reduces it compared to an unconditional prediction.** For example, without a stated intention (unconditional), a person in a living room could go anywhere. But with the high-level goal 'entertaining friends,' their future actions are strongly constrained to a set of social behaviors like offering drinks or sitting for a chat, rather than, say, going to bed.
>
> To enable models to learn the distribution of plausible motions that _still exists_ under a given intention, **we designed our data collection to explicitly capture this diversity**. We achieved this through two main strategies: (1) recruiting **diverse subjects** (varied gender, age, and physique) who performed actions based on **diverse scripts**, and (2) intentionally having participants **perform the same task multiple times**, resulting in a collection of different, plausible motion sequences for the exact same textual instruction. Our dataset is therefore **the first to provide the data that enables learning this one-to-many mapping** from a high-level intention to a distribution of plausible motions.
>
> &nbsp;
>
> ### **2. On the Validity of the Evaluation and Metrics**
>
> We agree with the reviewer that comparing to a single "ground truth" using error metrics like ADE/FDE is insufficient for this task and that a valid metric should evaluate whether the motion is "reasonable for the given intention."
>
> To better assess performance on this one-to-many prediction task, we incorporate probabilistic metrics like **MMADE and MMFDE** from **Appendix A.1 (L7-19)**. These metrics evaluate our K predicted motions ($p_k$) against a set of N plausible ground truth motions ($\hat{p_n}$). This set of plausible futures is identified by finding all motions in the dataset whose past input motion is similar (distance below a threshold) to the current input motion. A low score means the model's predictions do a good job of **covering the diverse range of plausible futures**. We define these metrics with the equations shown below. We believe **these metrics better capture the nature of our task**, and we will move these results to the main paper in the revised version.
>
> $$\text{MMADE@K} = \frac{1}{K} \sum_{k=1}^{K} \min_{n \in \{1, ..., N\}} \text{ADE}(p_k, \hat{p}_n)$$
>
> $$\text{MMFDE@K} = \frac{1}{K} \sum_{k=1}^{K} \min_{n \in \{1, ..., N\}} \text{FDE}(p_k, \hat{p}_n)$$
>
> Furthermore, to move beyond purely geometric evaluations and towards semantic "reasonableness," we introduce a new metric, **MotionCLIP Error (MCE)**, which measures the perceptual similarity between motions. We define it as:
>
> $$\text{MCE@K} = \min_{k \in \{1, ..., K\}} \lVert \text{MotionCLIP}(p_k) - \text{MotionCLIP}(\hat{p}) \rVert_2$$
>
> While MCE offers a step beyond purely geometric comparisons, we acknowledge it may not fully capture the high-level intention. This is because the pre-trained MotionCLIP model was trained on pairs of action descriptions and motions, not intentions and motions, and it does not incorporate scene context. For future work, **we envision developing truer intention-aware metrics** that leverage multi-modal large language models capable of grounding high-level textual intentions within specific 3D scene contexts. We believe **our dataset provides an ideal testbed for developing such crucial metrics.**
>
>
> To ensure our results are robust, we have re-run our experiments 5 times. Due to limited computational resources, we have completed this for the Self Intention Text, Pair Trajectory, and Pair Multimodal conditions. We will include the results for all conditions in the final manuscript. We present the mean and standard deviation of these runs in the table below. These results, which include the probabilistic metrics (MMADE, MMFDE, APD) and our proposed semantic metric (MCE), confirm the stability of our findings, as the **overall trends are consistent with the single-run** results presented in our original submission. **Pair Trajectory** excels at geometric accuracy, while **Self Intention Text** yields the highest diversity (APD). **Pair Multimodal** is highly competitive on both multi-modal coverage (MMADE/MMFDE) and semantic similarity (MCE), **demonstrating the value of combining high-level intent with other cues**. We believe this new, more robust evaluation framework using a **comprehensive suite of metrics provides a much more "valid" assessment** and is a significant contribution in itself.
>
> |Condition|ADE@1 ↓|FDE@1 ↓|MPJPE@1 ↓|ADE@20 ↓|FDE@20 ↓|MPJPE@20 ↓|
> |---|---|---|---|---|---|---|
> |Self Intention Text|1.483 ± 0.015|1.524 ± 0.016|1.543 ± 0.013|0.875 ± 0.006|0.771 ± 0.003|0.960 ± 0.006|
> |Pair Trajectory|1.193 ± 0.014|1.314 ± 0.012|1.316 ± 0.012|0.677 ± 0.004|0.634 ± 0.007|0.835 ± 0.003|
> |Pair Multimodal|1.384 ± 0.018|1.441 ± 0.023|1.471 ± 0.016|0.828 ± 0.009|0.719 ± 0.011|0.940 ± 0.007|
>
> | Condition           | MMADE@20 ↓    | MMFDE@20 ↓    | APD@20 ↑      | MCE@20 ↓      | Non-coll. ↑   | Contact ↑     |
> | ------------------- | ------------- | ------------- | ------------- | ------------- | ------------- | ------------- |
> | Self Intention Text | 1.241 ± 0.003 | 1.219 ± 0.004 | 5.161 ± 0.028 | 1.462 ± 0.005 | 0.999 ± 0.000 | 0.624 ± 0.002 |
> | Pair Trajectory     | 1.031 ± 0.003 | 1.090 ± 0.002 | 4.679 ± 0.011 | 1.230 ± 0.004 | 0.998 ± 0.000 | 0.781 ± 0.004 |
> | Pair Multimodal     | 1.169 ± 0.002 | 1.169 ± 0.002 | 4.658 ± 0.007 | 1.281 ± 0.005 | 0.998 ± 0.000 | 0.697 ± 0.010 |
>
> &nbsp;
>
> ### **3. On the Accessibility of the Dataset**
>
> We sincerely apologize for any issues with dataset accessibility. The conference-mandated Croissant format is new to us, and our file may not be configured correctly. For direct and reliable access, **we kindly ask you to use the Google Drive links provided in Appendix C (L82-87)**. These folders have not been modified since the original submission deadline. While the full dataset is large, we have also provided a **subset (approx. 10GB)** which should be much easier to inspect.
>
> The reviewer is correct that proper documentation and tools should have been provided. The data is organized by modality as follows:
>
> ```
> - Depth/
>   - Seq001/, Seq002/, ...
> - HumanMotion/
>   - Seq001/, Seq002/, ...
> - PointCloud/
>   - Seq001.ply, Seq002.ply, ...
> - RGB/
>   - Seq001/, Seq002/, ...
> - input_text.json
> ```
>
> For the final version, **we will provide comprehensive documentation and utility scripts, including a convenient PyTorch-style Dataset class** for easy data loading and processing, to ensure the dataset is truly accessible and useful for the community.
>
>
> ---
>
> &nbsp;
>
> We are confident that with these significant improvements—**a much more thorough and meaningful evaluation and a fully accessible and documented dataset**—our work now provides a strong and valuable resource for the community. We hope that these clarifications and revisions have addressed your primary concerns. Thank you again for the critical feedback that prompted these improvements.

---

### Official Review · Reviewer_GE5B · 2025-06-26

**Rating:** 4
**Confidence:** 3

**Summary:**

To address the limitations of previous work that primarily focuses on short-term kinematics and lacks explicit representations of human intention, this paper introduces a novel dataset named I2M (Intend-to-Move) for intention-aware long-term human motion modeling. In addition, the paper conducts a comprehensive analysis across eight conditioning features. Experimental results demonstrate that incorporating structured motion cues significantly improves prediction accuracy.

**Additional Feedback:**

I have no further comments; my main concerns lie in the limitations and weaknesses of the work.

**Dataset Code Accessibility:**

Yes

**Dataset Code Comments:**

The dataset code is provided.

**Ethical Considerations:**

No, there are no or only very minor ethics concerns

**Final Justification:**

The authors' response has addressed my concerns. Therefore, I keep my initial rating as "Borderline Accept".

**Limitations Weaknesses:**

1. The dataset does not clarify the number of subjects included. If all motions come from limited subjects, it raises concerns about the generalizability of motion patterns across individuals. Has the paper considered inter-subject variability?

2. The annotation procedure for high-level textual goals is not described in detail. It remains unclear how the temporal alignment between textual goals and corresponding actions is ensured.

3. The authors are encouraged to include more examples in the supplementary materials, or consider building a dataset page to better illustrate the data.

**Strengths Contributions:**

1. The proposed dataset targets intention-aware motion modeling. It comprises rich multimodal data, including SMPL pose sequences, multi-view RGB-D videos, scene point clouds, and high-level textual descriptions for goals.

2. The dataset is annotated with high-level intention labels, making it well-suited for studying the causal relationship between abstract goals and physical behavior.

3. The authors use the AffordMotion model as baseline, and evaluate the feature from different modalities, such as intension text, rgb images, trajectory, and semantic embeddings from MotionCLIP.

---

> ### Author Rebuttal · Authors · 2025-07-29
>
> We thank the reviewer for their time and valuable feedback. We are glad the reviewer recognized that our dataset **"targets intention-aware motion modeling"** with **"rich multimodal data"** and is **"well-suited for studying the causal relationship between abstract goals and physical behavior."**
>
> We believe their constructive comments will help us elevate the paper for the final version. We hope to clarify the main concerns, which we address point-by-point below.
>
> &nbsp;
> ### **1. Number of Subjects and Inter-subject Variability**
>
> We apologize if this was not sufficiently clear in our submission. As stated in **Section 3.4, "Dataset Statistics" (L217)**, our dataset includes **14 individuals (7 males, 7 females)**.
>
> To address the reviewer's important question about whether we **"considered inter-subject variability,"** we would like to provide more detail on our participant recruitment strategy. We recruited healthy adults, ensuring a **balanced distribution across both gender and age (20-50)**. Furthermore, within each gender, we recruited a balanced distribution of participants with **clothing sizes ranging from S to XL** to capture a variety of body shapes. We will add this more detailed, anonymized aggregate information to Section 3.4 in the revised manuscript.
>
> &nbsp;
> ### **2. Annotation Procedure for Textual Goals**
>
> We apologize that our annotation procedure was not described clearly enough and thank you for the opportunity to elaborate. Our method is a two-stage process to ensure accurate temporal alignment:
>
> 1. **Pre-collection Scripting**: Prior to data collection, we designed a high-level scenario script to guide the interaction. For example:
>
>     > ```
>     > - The scene opens with Person1 and Person2 standing at the entrance of the living room... They survey the room, taking in the mess.
>     >
>     > - Person1 takes initiative, stepping forward with the broom. They start sweeping the floor...
>     >
>     > - Person2, meanwhile, lays out a cleaning rag... and methodically begins wiping down surfaces...
>     > ```
>
> 2. **Post-collection Annotation**: After collection, we reviewed the synchronized video and motion data. We then manually annotated the precise start frame for each **intention (high-level textual goal)**. This resulted in a detailed, timestamped log of each participant's goals or evolving knowledge about the scene. For example, the final annotated data for a sequence includes:
>
>     > ```
>     > - frame_id: 0, subject: p1, text: The person plans to clean the room with her partner.
>     > - frame_id: 0, subject: p2, text: The person plans to clean the room with her partner.
>     > - frame_id: 0, subject: p1, text: The area under the carpet has not been cleaned for a long time.
>     > - ...
>     > - frame_id: 2817, subject: p2, text: The person discovered that there were magazines under the sofa.
>     > - ...
>     > ```
>
>
> This process ensures that each high-level textual goal is accurately aligned with the corresponding motion sequence. We will add this detailed explanation to the revised manuscript.
>
> &nbsp;
> ### **3. Including More Examples**
>
> We thank the reviewer for this constructive suggestion. We agree that providing more visualizations is essential for showcasing the dataset's value. In the revised version, we will add more extensive visualizations, including **new visualizations of the rich, multi-modal data in our dataset** (e.g., sample motion sequences and scenes) and **qualitative results of our model's performance**. Furthermore, we will **create a dedicated dataset webpage** featuring more examples, videos, and detailed documentation to better serve the research community.
>
> ---
>
> &nbsp;
>
> We hope these clarifications and our planned revisions directly address the reviewer's main concerns. We are grateful for the feedback, which will help us significantly improve the clarity and completeness of our paper.

---

> > ### Comment · Reviewer_GE5B · 2025-08-04
> > **Response to Rebuttal**
> >
> > The authors' response has addressed my concerns. Therefore, I keep my initial rating as "Borderline Accept".

---

> > > ### Author Response · Authors · 2025-08-04
> > >
> > > Thank you for taking the time to read our rebuttal and for your follow-up comment. We are glad to hear that our response has addressed your concerns. We appreciate your valuable feedback throughout this process.

---

### Official Review · Reviewer_ZaXg · 2025-06-30

**Rating:** 5
**Confidence:** 4

**Summary:**

This paper introduces Intend to Move (I2M), a new large-scale dataset for intention-aware human motion prediction in dynamic real-world home environments. I2M contains 10.1 hours of SMPL-based 3D human motion across 215 long-term sequences, annotated with multi-view RGB-D videos, 3D scene point clouds, and natural language intention descriptions. The dataset emphasizes human intention, human-scene interaction, and multi-human social behavior. To benchmark I2M, the authors adapt the AffordMotion method by modifying its diffusion-based architecture to better accommodate the unique properties of the I2M dataset. They conduct a comprehensive evaluation of the predictive power of various input modalities, including trajectory, MotionCLIP embeddings, RGB images, and intention text.

**Additional Feedback:**

1.	This is a promising dataset that focuses on the task of intention-conditioned human motion prediction. Beyond the tasks defined in the paper, I believe this dataset could also serve as a valuable resource for related areas such as action planning and motion generation.
2.	Exploring a broader range of behavioral diversity would further enhance the value of this dataset. In my opinion, factors such as age, gender, and personality traits can lead to significant variations in motion patterns. A dataset dominated by a single demographic profile may introduce bias and limit the model’s ability to generalize to real-world scenarios. Future efforts to capture more diverse motion behaviors would be highly beneficial to the research community.

**Dataset Code Accessibility:**

Yes

**Dataset Code Comments:**

The authors have indicated that both dataset and code will be released, with URLs provided. The paper also offers detailed documentation of data collection, annotation, and processing pipelines (Sections 3.2 and 3.3), ensuring good reproducibility.

**Ethical Considerations:**

No, there are no or only very minor ethics concerns

**Final Justification:**

Thanks for the replies from the authors. To remain consistent with the original decision, I believe this paper can be accepted.

**Limitations Weaknesses:**

1.	Limited Subject Demographic Information:
The authors only report the gender and the number of participants in Section 3.4, without providing details about participants’ age, body types, physical conditions, or other demographic attributes. However, human motion is influenced not only by intention but also by individual physical characteristics. The lack of such information may affect the interpretation and generalizability of the dataset.
2.	Lack of Statistical Analysis on Experimental Results:
The paper does not report error statistics or variance analysis based on multiple runs. This omission makes it difficult to assess the stability and robustness of the experimental findings.
Lack of Intention-Aware and Semantic Evaluation Metrics
3.	Lack of Intention-Aware and Semantic Evaluation Metrics
The evaluation metrics used in this paper are primarily low-level physical measures, including Average Displacement Error (ADE), Final Displacement Error (FDE) and Mean Per Joint Position Error (MPJPE). While these metrics are well-suited for measuring geometric fidelity in motion prediction tasks, they are insufficient for evaluating whether the predicted motion successfully fulfills the intended goal. For example, a predicted trajectory might exhibit low ADE or FDE by following a spatially close path, yet fail to interact with relevant scene elements (e.g., walking toward the kitchen without engaging with any target objects or areas), thereby not accomplishing the intended action. Moreover, such simple distance-based metrics do not capture the semantic correctness, intention consistency, or plausibility of the predicted motion. This limitation could potentially mislead the community’s interpretation of model performance, especially in intention-driven tasks where semantic goal achievement is a more meaningful success criterion than purely spatial proximity. I encourage the authors to consider incorporating semantic-level evaluation metrics or task-specific success rates in future work to provide a more comprehensive and intention-aware assessment.

**Strengths Contributions:**

1.	Filling a Dataset Gap:
I2M fills an important gap in human motion research by collecting a large-scale, intention-driven multimodal dataset in real-world environments. It addresses several limitations present in existing human motion datasets, particularly in terms of intention modeling and scene-aware interactions. (Sec 2)
2.	Novel Dataset Design:
The dataset captures human motion in realistic home settings, featuring diverse types of human-human and human-object interactions. This significantly enhances the ecological validity of the data. Moreover, I2M provides multiple synchronized modalities—including 3D motion, multi-view RGB-D, 3D scene point clouds, and natural language descriptions—which enables research on cross-modal learning and intention-grounded tasks. (Sec3)
3.	Detailed and Rigorous Data Description:
The paper presents a thorough and carefully documented data collection and cleaning pipeline, ensuring high data quality and reproducibility. (Sec 3)
4.	Benchmark Experiments:
The authors conduct a baseline evaluation by adapting and fine-tuning an existing diffusion-based model. Through experiments under various input modality configurations, the paper provides valuable insights into the relationships and contributions of different modalities. These benchmark results can serve as an important reference for future research on intention-aware motion modeling. (Sec 4)

---

> ### Author Rebuttal · Authors · 2025-07-29
>
> We sincerely thank the reviewer for their thorough, insightful, and highly supportive review of our work. We are thrilled that the reviewer recognized our key contributions, noting that our dataset **"fills an important gap"** and has a **"novel dataset design."** We are especially grateful for the feedback that this is **"a promising dataset"** that could be a **"valuable resource for related areas such as action planning and motion generation."** We agree this is an excellent suggestion and will add this point to our discussion on future directions in the revised manuscript.
>
> We appreciate the constructive feedback, which we believe will significantly strengthen our paper. We address each point below.
>
> &nbsp;
> ### **1. Limited Subject Demographic Information**
>
> Thank you for raising this important point regarding generalizability. To provide a clearer picture of our participants, we offer the following anonymized details: We recruited healthy adults, ensuring a **balanced distribution across both gender and age (20-50)**. Furthermore, within each gender, we recruited a balanced distribution of participants with **clothing sizes ranging from S to XL** to capture a variety of body shapes. We will add this anonymized aggregate information to Section 3.4 in the revised manuscript.
>
> **Our current dataset already captures a degree of diversity** through our balanced recruitment. That said, we completely agree with your "Additional Feedback" that "exploring a broader range of behavioral diversity" by including factors like a wider age range and personality traits would further enhance its value. We will add a discussion on this as a critical direction for future data collection efforts.
>
> &nbsp;
> ### **2. Lack of Intention-Aware and Semantic Evaluation Metrics**
>
> We are extremely grateful for this insightful and detailed critique. We completely agree that low-level physical metrics like ADE and FDE are insufficient for evaluating whether a predicted motion successfully fulfills the intended goal.
>
> Motivated by this exact concern, we have introduced a new semantic-level metric, **MotionCLIP Error (MCE)**, which measures the perceptual similarity between the predicted and ground truth motions. We define it as the minimum L2 distance between the MotionCLIP embeddings of the K predicted motions ($p_k$​) and the ground truth motion ($\hat{p}$​), as shown in the formula below. The results for this new MCE metric are included in the comprehensive statistical analysis presented in the following section.
>
> $$\text{MCE@K} = \min_{k \in \{1, ..., K\}} \lVert \text{MotionCLIP}(p_k) - \text{MotionCLIP}(\hat{p}) \rVert_2$$
>
>
> While MCE offers a step beyond purely geometric comparisons, we acknowledge it may not fully capture the high-level intention. This is because the pre-trained MotionCLIP model was trained on pairs of action descriptions and motions, not intentions and motions, and it does not incorporate scene context. For future work, **we envision developing truer intention-aware metrics** that leverage multi-modal large language models capable of grounding high-level textual intentions within specific 3D scene contexts. We believe **our dataset provides an ideal testbed for developing such crucial metrics.**
>
> &nbsp;
> ### **3. Lack of Statistical Analysis on Experimental Results**
>
> This is a crucial point, and we thank the reviewer for it. To ensure the stability and robustness of our experimental findings, we have re-run our experiments 5 times with different random seeds. Below, we present the mean and standard deviation of these runs. **The small standard deviations across all metrics confirm the stability of the results reported in our original manuscript.**  Due to limited computational resources, we have completed this for the Self Intention Text, Pair Trajectory, and Pair Multimodal conditions. We will include the results for all conditions in the final manuscript.
>
> The **Pair Trajectory** condition shows strong performance on geometric accuracy (ADE/FDE/MPJPE). However, the other conditions are more competitive on the probabilistic and semantic metrics. For instance, **Self Intention Text** yields the highest diversity (APD score), while **Pair Multimodal** achieves strong multi-modal coverage (MMADE/MMFDE). Additionally, **Pair Multimodal also remains highly competitive on semantic similarity (MCE)**, closely rivaling the Pair Trajectory condition. This suggests that while low-level cues are effective for predicting a single path, **high-level semantic inputs play a key role for capturing the multi-modal and diverse nature of human motion**, indicating a promising direction for future research.
>
> |Condition|ADE@1 ↓|FDE@1 ↓|MPJPE@1 ↓|ADE@20 ↓|FDE@20 ↓|MPJPE@20 ↓|
> |---|---|---|---|---|---|---|
> |Self Intention Text|1.483 ± 0.015|1.524 ± 0.016|1.543 ± 0.013|0.875 ± 0.006|0.771 ± 0.003|0.960 ± 0.006|
> |Pair Trajectory|1.193 ± 0.014|1.314 ± 0.012|1.316 ± 0.012|0.677 ± 0.004|0.634 ± 0.007|0.835 ± 0.003|
> |Pair Multimodal|1.384 ± 0.018|1.441 ± 0.023|1.471 ± 0.016|0.828 ± 0.009|0.719 ± 0.011|0.940 ± 0.007|
>
> | Condition           | MMADE@20 ↓    | MMFDE@20 ↓    | APD@20 ↑      | MCE@20 ↓      | Non-coll. ↑   | Contact ↑     |
> | ------------------- | ------------- | ------------- | ------------- | ------------- | ------------- | ------------- |
> | Self Intention Text | 1.241 ± 0.003 | 1.219 ± 0.004 | 5.161 ± 0.028 | 1.462 ± 0.005 | 0.999 ± 0.000 | 0.624 ± 0.002 |
> | Pair Trajectory     | 1.031 ± 0.003 | 1.090 ± 0.002 | 4.679 ± 0.011 | 1.230 ± 0.004 | 0.998 ± 0.000 | 0.781 ± 0.004 |
> | Pair Multimodal     | 1.169 ± 0.002 | 1.169 ± 0.002 | 4.658 ± 0.007 | 1.281 ± 0.005 | 0.998 ± 0.000 | 0.697 ± 0.010 |
>
>
> ---
>
> &nbsp;
>
> Thank you once again for the detailed, thoughtful, and encouraging review. Your feedback has been invaluable in helping us strengthen our work.

---

### Official Review · Reviewer_R4Wh · 2025-07-01

**Rating:** 5
**Confidence:** 5

**Summary:**

This paper proposes a dataset for intention and scene aware human motion prediction.

**Dataset Code Accessibility:**

Yes

**Dataset Code Comments:**

The dataset provides the accessible URL. And the data is available.

**Ethical Considerations:**

No, there are no or only very minor ethics concerns

**Final Justification:**

This paper introduces a novel dataset for human motion prediction, which incorporates motion intention, interaction, and scene. The concerns in the initial review regarding human representation and experimental details have been addressed in the rebuttal. Therefore, my final recommendation is Accept.

**Limitations Weaknesses:**

Thanks to the author for providing the dataset. It is important for intention-based human motion generation. However, there are still some concerns:
1. About SMPL model. In this work, the human is represented by SMPL model. However, a more natural human interaction should be achieved through SMPL-X model, which includes hand pose and shape. Because the interaction of hands is also important for human interaction, which implies more detailed intentions.

2. The experimental section is rather simple. More details of the experiments should be provided. For instance, the selection basis and calculation of evaluation metrics, as well as the visualization of prediction results, etc.

**Strengths Contributions:**

The strength of this work is as follows:
1. It clearly explains the motivation. In human interaction, intention is very important, and this motivation is of great value.

2. The dataset contains rich information, including intention-driven motion, interactions, and scenes.

3. It is interesting that the work classifies human intentions into four categories.

4. The dataset provides detailed information on the data collection process and annotation.

---

> ### Author Rebuttal · Authors · 2025-07-29
>
> We sincerely thank the reviewer for their valuable time and very positive assessment of our work. We are greatly encouraged that the reviewer recognized that our work **"clearly explains the motivation,"** that the **"dataset contains rich information,"** and that it **"provides detailed information on the data collection process and annotation."**
>
> We address the reviewer's concerns below.
>
> &nbsp;
> ### **1. Regarding the use of SMPL vs. SMPL-X**
>
> We thank the reviewer for this insightful point and agree that hand motion is crucial for detailed intention understanding. We used the SMPL model as our primary focus was on the link between high-level intentions and overall body motion, for which SMPL is a well-established standard.
>
> For future work, we plan to build upon this foundation. We intend to create a new, richer dataset that **includes expressive hand motions**, likely using SMPL-X, to explore these finer-grained interactions.
>
> &nbsp;
> ### **2. Regarding the details of the experimental section**
>
> We apologize that our manuscript was not sufficiently clear regarding the experimental details and appreciate the opportunity to clarify and extend our evaluation.
>
> #### **(i) Selection and Calculation of Metrics**
> Our rationale for the choice of metrics is briefly mentioned in **Section 4.2 (L263-266)**. To clarify, we adopted **ADE, FDE, and MPJPE** as they are standard benchmarks in the human motion prediction field. We also followed our baseline, AffordMotion, in using the **Contact and Non-collision** metrics to evaluate the physical plausibility of the predicted motions. To assess the model's ability to handle the inherent ambiguity of human motion, we also used **APD** to measure diversity and **MMADE/MMFDE** to evaluate coverage of multiple plausible futures in the **Appendix A.2 (L20-55)**. The detailed descriptions and mathematical formulations for all these metrics are provided in **Appendix A.1 (L7-19)**. We sincerely apologize for not explicitly referencing this appendix in the main text and will add a clear pointer in the revised version.
>
> #### **(ii) Introduction of MotionCLIP Error**
> To address the reviewer's valid point and to move beyond purely geometric evaluations, we introduce a new semantic-level metric, MotionCLIP Error (MCE), which measures the perceptual similarity between motions. We define it as the minimum L2 distance between the MotionCLIP embeddings of the K predicted motions ($p_k$​) and the ground truth motion ($\hat{p}$​):
>
> $$\text{MCE@K} = \min_{k \in \{1, ..., K\}} \lVert \text{MotionCLIP}(p_k) - \text{MotionCLIP}(\hat{p}) \rVert_2$$
>
> While this provides a valuable semantic evaluation, we acknowledge that it may not fully capture the high-level intention behind an action. This is because the pre-trained MotionCLIP model was trained on pairs of action descriptions and motions, not intentions and motions, and it does not incorporate scene context. For future work, **we envision developing truer intention-aware metrics** that leverage multi-modal large language models capable of grounding high-level textual intentions within specific 3D scene contexts. We believe **our dataset provides an ideal testbed for developing such crucial metrics.**
>
> #### **(iii) Robustness of Results and Analysis**
> To ensure the stability of our findings, we have re-run our experiments 5 times. Due to limited computational resources, we have completed this for the Self Intention Text, Pair Trajectory, and Pair Multimodal conditions. We will include the results for all conditions in the final manuscript. Below, we present the mean and standard deviation of these runs. **The overall trends are consistent with the single-run** results presented in our original submission, confirming the stability of our findings. The results offer several key insights, revealing the distinct strengths of each approach. The **Pair Trajectory** condition excels at geometric accuracy (ADE/FDE/MPJPE), while **Self Intention Text** yields the highest diversity (APD). Notably, **Pair Multimodal** not only achieves strong multi-modal coverage (MMADE/MMFDE) but also remains highly competitive on semantic similarity (MCE), closely rivaling the Pair Trajectory condition. This suggests that while low-level cues are effective for predicting a single path, **high-level semantic inputs play a key role for capturing the multi-modal and diverse nature of human motion**, indicating a promising direction for future research.
>
>
> |Condition|ADE@1 ↓|FDE@1 ↓|MPJPE@1 ↓|ADE@20 ↓|FDE@20 ↓|MPJPE@20 ↓|
> |---|---|---|---|---|---|---|
> |Self Intention Text|1.483 ± 0.015|1.524 ± 0.016|1.543 ± 0.013|0.875 ± 0.006|0.771 ± 0.003|0.960 ± 0.006|
> |Pair Trajectory|1.193 ± 0.014|1.314 ± 0.012|1.316 ± 0.012|0.677 ± 0.004|0.634 ± 0.007|0.835 ± 0.003|
> |Pair Multimodal|1.384 ± 0.018|1.441 ± 0.023|1.471 ± 0.016|0.828 ± 0.009|0.719 ± 0.011|0.940 ± 0.007|
>
> | Condition           | MMADE@20 ↓    | MMFDE@20 ↓    | APD@20 ↑      | MCE@20 ↓      | Non-coll. ↑   | Contact ↑     |
> | ------------------- | ------------- | ------------- | ------------- | ------------- | ------------- | ------------- |
> | Self Intention Text | 1.241 ± 0.003 | 1.219 ± 0.004 | 5.161 ± 0.028 | 1.462 ± 0.005 | 0.999 ± 0.000 | 0.624 ± 0.002 |
> | Pair Trajectory     | 1.031 ± 0.003 | 1.090 ± 0.002 | 4.679 ± 0.011 | 1.230 ± 0.004 | 0.998 ± 0.000 | 0.781 ± 0.004 |
> | Pair Multimodal     | 1.169 ± 0.002 | 1.169 ± 0.002 | 4.658 ± 0.007 | 1.281 ± 0.005 | 0.998 ± 0.000 | 0.697 ± 0.010 |
>
> #### **(iv) Qualitative Results:**
> We have provided qualitative results including visualizations in **Appendix A.3 (L56-73)**. To better illustrate our model's performance and strengthen the experimental section, we will move the qualitative results to the main paper in the revised version.
>
>
> ---
>
> &nbsp;
>
> We hope these clarifications and extensive new results address the reviewer's concerns. We are very grateful for the constructive feedback, which will help us significantly improve our paper.

---

> > ### Comment · Reviewer_R4Wh · 2025-08-05
> > **Response to Rebuttal**
> >
> > Thanks for the work in the rebuttal. My concerns have been addressed. I keep my vote.

---

> > > ### Author Response · Authors · 2025-08-05
> > >
> > > Thank you for taking the time to review our rebuttal and for your positive follow-up. We are very pleased to hear that we have addressed your concerns. We sincerely appreciate your valuable feedback and strong support for our work throughout this process.

---

### Note · Authors · 2025-08-12

To the Area Chair and Reviewers,

We sincerely thank you for the thorough and constructive review process. We were greatly encouraged by the positive feedback, highlighted by comments that our work **"fills an important gap"** with **"rich multimodal data,"** is **"clearly different from previous ones,"** and serves as a **"valuable resource for related areas such as action planning and motion generation."**

In response to the invaluable feedback, we have made the following significant improvements:

1. **More Robust Evaluation:** Conducted new experiments (5 runs) to confirm the stability of our results (in response to R-ZaXg).
2. **More Intention-Aware Metrics:** Introduced new semantic (MCE) and probabilistic (MMADE/MMFDE) metrics to provide a more intention-aware assessment (in response to R-R4Wh, R-ZaXg and R-nZdu).
3. **Improved Dataset Accessibility:** Committed to providing comprehensive documentation and utility scripts to resolve the accessibility concerns (in response to R-nZdu).
4. **Enhanced Clarity:** Clarified our methodology for participant selection and annotation. We will also add more qualitative results to the main paper, supplementary materials, and a new project page (in response to R-R4Wh and R-GE5B).

We believe these improvements have addressed the primary concerns and strengthened our contribution. We are sincerely grateful for the thoughtful and constructive feedback, which has been invaluable in helping us substantially improve our paper.

---

### Decision · Program_Chairs · 2025-09-18

**Decision:**

Accept (poster)

**Comment:**

All the reviews are positive, i.e., BA, BA, A, and A. The reviewers recognize the strong points of this work as follows:

- The new I2M dataset is a significant contribution, as it is a large-scale, multimodal dataset that captures intention-driven human motion in realistic home settings. It is rich in information, including synchronized 3D motion, scene data, and natural language descriptions.
- The paper's rigorous methodology includes a detailed data collection process and valuable benchmark experiments. This work provides a strong foundation for future research on intention-aware motion modeling, with its findings and data directly serving the community.

After reading the rebuttal and author-reviewer discussions, the reviewers believe that this paper is ready for the presentation at NeurIPS. Thus, the decision is Accept.
Please read the final review comments to prepare the camera-ready manuscript because some reviewers still have some concers.